# Polyphenol Profile and Biological Activity Comparisons of Different Parts of *Astragalus macrocephalus* subsp. *finitimus* from Turkey

**DOI:** 10.3390/biology9080231

**Published:** 2020-08-17

**Authors:** Cengiz Sarikurkcu, Gokhan Zengin

**Affiliations:** 1Department of Analytical Chemistry, Faculty of Pharmacy, Afyonkarahisar Health Sciences University, Afyonkarahisar 03030, Turkey; sarikurkcu@gmail.com; 2Department of Biology, Science Faculty, Selcuk University, Campus, Konya 42130, Turkey

**Keywords:** *astragalus*, antioxidant, α-amylase, hyperoside, bioactive compounds

## Abstract

The members of the genus *Astragalus* have great interest as traditional drugs in several folk systems including Turkey. In this sense, the present paper was aimed to explore the biological properties and chemical profiles of different parts (aerial parts, leaves, flowers, stems, and roots) of *A. macrocephalus* subsp. *finitimus*. Antioxidant (radical quenching, reducing power, and metal chelating) and enzyme inhibitory (α-amylase and tyrosinase) effects were investigated for biological properties. Regarding chemical profiles, individual phenolic compounds were detected by LC-MS, as well as total amounts. The leaves extract exhibited the strongest antioxidant abilities when compared with other parts. However, flowers extract had the best metal chelating ability. Hyperoside, apigenin, p-coumaric, and ferulic acids were identified as main compounds in the tested parts. Regarding enzyme inhibitory properties, tyrosinase inhibitory effects varied from IC_50_: 1.02 to 1.41 mg/mL. In addition, the best amylase inhibition effect was observed by leaves (3.36 mg/mL), followed by aerial parts, roots, stems, and flowers. As a result, from multivariate analysis, the tested parts were classified in three cluster. Summing up the results, it can be concluded that *A. macrocephalus* subsp. *finitimus* could be a precious source of natural bioactive agents in pharmaceutical, nutraceutical, and cosmeceutical applications.

## 1. Introduction

Since the beginning of the last century, scientists have been focused on the biological and chemical properties of plants with ethnobotanical evidence [1,2,3]. From their studies, several important compounds have been introduced. As a springboard, the ethnobotanical records in ancient times indicated that *Artemisia annua* had great potential against malaria. In the light of this information, Japanese and Chinese scientist have isolated one sesquiterpene (artemisinin) from this plant to combat malaria, which they won the Nobel Prize for in 2015 [4,5]. In this sense, traditional and scientific data have to combine for further applications. Turkey has significant ethnobotanical data, with remarkable floristic features (about 12,000 plants) [6]. However, most of them have been scarcely investigated. Thus, the uninvestigated plants could be considered a treasure for pharmaceutical and medicinal applications.

In the last decade, plant secondary metabolites, especially phenolic compounds, have been gaining interest in the scientific platform. These compounds contain one or more hydroxyl groups and they have good hydrogen/electron donating abilities. Thus, these compounds are considered as main contributors to antioxidant properties. Additionally, these compounds have a broad spectrum of biological activities such as antimicrobial, anti-inflammatory, and anti-cancer [7,8].

The genus *Astragalus* is one of the biggest genera in the family Fabaceae and is represented by more than 2500 species [9]. The genus also contains 478 taxa in Turkey and it has many endemic species (202, endemism ratio: 42%) to Turkey [10]. Regarding folk medicinal uses, the genus is traditionally used for several purposes. For example, *A. gumnifer* and *A. longifolius* roots are used to treat diabetes mellitus [11]. Additionally, *A. aureus* and *A. brachylcalyx* are used against stomachache and sore throat [12]. In addition, *A. lamarckii* for ulcer [13]; *A. cephalotes* var. *brevicalyx* for wound healing [14] and *A. tmoleus* for abdominal pain and toothache [15]. From the light of these ethnobotanical records, several biological and chemical studies were performed on the members of the genus [16,17,18,19,20,21,22]. In the chemical studies, some biologically-active compounds, including hyperoside, apigenin, kaempferol, and naringenin, were detected [9]. However, to the authors best knowledge, very few publications can be found biological properties of *Astragalus microcephalus* [23,24,25]. *A. microcephalus* is a stout and erect perennial plant (50–100 cm). Leaves are lanceolate and narrowly elliptic. Inflorescence is 3.5–5 mm diameter and contains 30–50 sessile flowers. Calyx is 15–18 mm and tubular-campanulate. Corolla is 18–35 mm and deep yellow [26]. In the current work, we aimed to examine biological properties (antioxidant and enzyme inhibitory effect) and chemical composition (total and individual phenolic compounds) of different parts (aerial parts, leaves, flowers, stems and roots) of *A. macrocephalus* subsp. *finitimus*.

## 2. Materials and Methods

### 2.1. Plant Material and Solvent Extraction

*Astragalus macrocephalus* Willd. subsp. *finitimus* (Bunge) Chamberlein (Fabaceae) were collected from Sucati village, Gurun, Sivas-Turkey on 23 June, 2019 (1351 m, 38°43′15.06” N 37°21′43.22” E), authenticated by Olcay Ceylan, and deposited (AD-1518) at the Department of Biology, Mugla Sıtkı Koçman University (Mugla, Aegean, Turkey). The plant was collected in the flowering season and the aerial parts do not contain fruit and seeds. The plant was firstly divided into different parts (aerial parts (as mix leaves, flowers, and stems) roots, leaves, flowers, and stems). The plant materials were dried in a shaded and well-ventilated environment (about 10 days) and were powdered in a laboratory mill. After powdering process, the plant materials were used to obtain extracts in the same week.

The methanol extracts from different parts of *A. macrocephalus* subsp. *finitimus* were prepared by maceration for 24 h. Five grams of different parts (aerial parts, roots, leaves, flowers, and stems) were mixed with 100 mL of solvent (the ratio of solid/solvent: 1:20) and agitation was set to 150 rpm in dark environment at room temperature. All of the extracts were stored at +4 °C until analyzed after concentrating the methanol extracts under reduced pressure. Extraction yields were given in Table 1.

### 2.2. Total Flavonoid and Phenolic Contents

To obtain total level of phenolic (TPC) and flavonoid content (TFC) in the extracts, colorimetric assays were used as described in our previous paper [27]. Gallic acid (GAE) and quercetin (QE) were used as standards, respectively. Please see the Appendix A for the details.

### 2.3. Liquid Chromatography–Electrospray Tandem Mass Spectrometry (LC–ESI–MS/MS) Analysis

To determine chemical compositions in the extracts, we used an Agilent Technologies 1260 Infinity liquid chromatography system (Santa Clara, CA, USA) hyphenated to a 6420 Triple Quad mass spectrometer on which a chromatographic separation on a Poroshell 120 EC-C18 (100 mm × 4.6 mm I.D., 2.7 μm) column [28]. All analytical and chromatographic details are given in the Appendix A. The different analytes were identified by means of their retention times, mass spectra, and tandem mass spectra. Specifically, quantitative analyses were performed using a specific MRM transition for each analyte. Analytical parameters and chromatograms are given in Appendix A.

### 2.4. Biological Activity

Antioxidant properties of these extracts were detected by several assays including DPPH radical [29] ABTS^+^ free radical scavenging [30], cupric ion (CUPRAC) and ferric ion (FRAP) reducing power [31,32], phosphomolybdenum method [33] and ferrous ion chelating [34]. The antioxidant properties were evaluated by IC_50_ values (the half inhibitory concentration). The IC_50_ values were calculated from the graph of percentage (ABTS^+^, DPPH and metal chelating) against the concentration of the extracts. IC_50_ values for other assays (reducing power and phosphomolybdenum) reflect that the concentration at which absorbance is 0.5. For this purpose, we used the graph of absorbance against the concentration of the extracts. Trolox (TE) and Ethylenediaminetetraacetic acid (disodium salt) (EDTA)) were used as positive controls. In addition, the results were expressed as equivalents of these standards.

The key enzymes inhibition activity of the extracts against tyrosinase, and α-amylase were measured using the protocols as published by [35]. The enzyme inhibition abilities were evaluated by IC_50_ values. IC_50_ values calculated as antioxidant assays and we used a graph between concentration and percentage of enzyme inhibition. Standard enzyme inhibitors (Kojic acid (KAE) for tyrosinase and acarbose (ACE) for α-amylase) were used as positive control and also, the results were expressed as equivalents of these standards. The details for experimental methods are given in the Appendix A.

### 2.5. Statistical Analysis

Obtained results were given as mean ± standard deviation (SD) and the results were evaluated by ANOVA assay (with Tukey’s test, significant value: *p* < 0.05). Principal component analysis (PCA) and hierarchical clustered analysis (HCA) were applied to the experimental data under FactoMineR (Factor Analysis and Data Mining with R) package (R Core Team, Vienna, Austria). The antioxidant activities of the extracts were analyzed using various methods. As is well known, each of the antioxidant activity methods has a different mechanism of action on the extracts. Therefore, it is not possible to directly compare the results obtained with each other. Relative antioxidant capacity (RACI) index values were calculated to make the results comparable, and the correlation between the results obtained from each test and RACI values were presented separately [36]. The RACI values of the samples were determined for each test by dividing into standard deviation after subtracting these mean values from the raw data. Total RACI values were calculated by averaging the RACI values obtained from all antioxidant tests of the relevant sample (including phenolic and flavonoid).

## 3. Results and Discussion

### 3.1. Phytochemical Composition

The amounts of total phenolics and flavonoids in the tested extracts were affected by plant parts used. As shown in Table 1, the highest levels of phenolics and flavonoids were determined in the leaves extract (37.68 mg GAE/g and 39.23 mg QE/g). Flowers (6.96 mg GAE/g) and roots (6.03 mg QE/g) had the lowest level of total phenolics and flavonoids, respectively. Several studies reported different levels of these compounds in the members of the genus *Astragalus* [16,37,38,39]. Observed differences may be linked with geographical, environmental, and climatic conditions as well as plant parts [37,40,41,42]. In addition, recent studies indicated that the colorimetric methods had several drawbacks and these methods could not reflect accurate levels of these compounds in plant extracts [43,44]. Hence, chromatographic methods such as HPLC or LC-MS are required to provide certain data. In this context, the extracts were analyzed by LC-MS and the results are given in Table 2. Hyperoside, p-coumaric and ferulic acids and apigenin were identified as main compounds in the tested extracts. The level of hyperoside varied from 2.90 (in roots) to 1828.94 (in leaves) µg/g extract. The highest level of p-coumaric acid was detected in flowers extract with 146.78 µg/g extract. The main compounds in the extracts exhibited significant biological activities in earlier studies. For example, hyperoside is a main compound in the genus *Hypericum* and this compound exhibits promising biological abilities [45,46,47]. Additionally, similar properties were also reported for p-coumaric acid [48], apigenin [49] and ferulic acid [50,51]. From this point, observed biological activities of *A. macrocephalus* subsp. *finitimus* extracts might be linked to the presence of these compounds.

### 3.2. Antioxidant Properties

Oxidative stress is the main etiological factor for the progression of several chronic and degenerative diseases such as Alzheimer’s disease, cancer, and cardiovascular diseases. Thus, the balance between the production of free radicals and the endogenous antioxidant defense system plays a pivotal role in healthy physiological function [52]. At this point, we need to support the defense system with dietary antioxidants. Plants are the main sources of the dietary antioxidants and several studies have reported a negative association between the consumption of plants and the frequency of these diseases [53,54,55]. In the present study, to evaluate the antioxidant effects of *A. macrocephalus* subsp. *finitumus* extracts, several chemical methods were performed, and their results are shown in Table 3. We used IC_50_ values and standard equivalents (trolox (TE) and EDTA (EDTAE)) to express antioxidant abilities. Based on Table 3, the strongest antioxidant abilities were detected in leaves extracts. For example, the lowest IC50 values were detected in the leaves extract for radical scavenging (ABTS and DPPH) and to reduce power (FRAP, CUPRAC and phosphomolybdenum). Observed antioxidant effects for leaves extract could be explained with the high level of phenolics and we obtained a good correlation between these parameters Table 4. In accordance with our findings, several researchers reported a positive correlation between total phenolic content and antioxidant properties. Interestingly, the metal chelating abilities of the tested extracts can be ranked as flowers>stems>roots>aerial parts>leaves. In addition, a negative relationship was observed between total bioactive compounds (phenolics and flavonoids) and metal chelating ability. Taken together, we could imply that observed findings could be linked with the presence of non-phenolic chelators such as peptides, polysaccharides, and ascorbic acid. In earlier studies, several authors reported antioxidant properties of some *Astagalus* species such as *A. ponticus* [16], *A. lagurus* [56], *A. spruneri* [57], *A. membranaceus* [58,59]. With this in mind, the members of the genus *Astragalus* could be considered as valuable sources of natural antioxidants.

Several researchers suggested that only one method is not enough to evaluate antioxidant abilities of plant extracts and thus, multiple methods including different mechanisms are required to obtain a full antioxidant picture. [52,60]. However, different expression methods have been observed in these different methods. With this fact, any comparison between results might be unreasonable and sometimes impossible. Thus, relative antioxidant capacity index (RACI) has been developed by some researchers to obtain an accurate comparison between studies [36,61]. In the present study, we calculated the relative antioxidant capacity index for each part in Figure 1 and each method in Figure 2. Clearly, among the tested plant parts, the leaves had the strongest antioxidant ability, followed by aerial parts, stems, flowers and, roots. As shown in Figure 2, with one exception (metal chelation), the leaves exhibited the best ability in the methods performed. This fact also was confirmed by correlation analysis. The contradictory results from metal chelating assays might be explained with the presence of non-phenolic chelators such as polysaccharides, peptides, and sulphates. This approach was observed in earlier studies [62,63].

### 3.3. Inhibitory Effects on Amylase and Tyrosinase

Enzyme inhibition theory is one of the most important strategies to combat global health problems including Alzheimer’s disease and diabetes. In theory, some enzymes are targets to alleviate observed symptoms in the diseases [64]. For example, amylase is one of the main enzymes in the carbohydrate catabolism and it hydrolyzes α (1,4) glycosidic bonds in the starch. Thus, the inhibition of amylase can control the postprandial blood glucose level [65]. Additionally, tyrosinase is a key enzyme in the synthesis of melanin and its inhibition can reduce the symptoms of hyperpigmentation problems [66]. Thus, several compounds (acarbose for amylase and kojic acid for tyrosinase) have been developed as enzyme inhibitors in pharmaceutical industries. However, most of them have serious side effects such as gastrointestinal disorders and toxicity [67,68,69]. In this sense, natural substances prefer as enzyme inhibitors against synthetic ones.

Amylase and tyrosinase inhibition of *A. macrocephalus* subsp. *finitimus* extracts were investigated and the results are reported in Table 5. Similar to antioxidant assays results, the best inhibitory ability was detected in leaves extract (IC50: 3.36 mg/mL for amylase and 1.02 mg/mL for tyrosinase). In addition, the flowers exhibited the weakest inhibitory activities (IC_50_: 4.94 mg/mL for amylase and 1.41 mg/mL for tyrosinase). The findings could be related with chemical profiles of the tested extracts and some compounds in extracts such as hyperoside [70,71], ferulic acid [72,73], and apigenin [74,75] have been reported as inhibitory agents in earlier studies. A moderate positive correlation was also observed between total phenolic content and the enzyme inhibitory abilities Table 4. As far as we know, no information on the enzyme inhibitory effect of *A. macrocephalus* is present. Therefore, our results could provide new information on the biological activity poof for the genus *Astragalus*. At this point, *A. microcephalus* could be considered as a valuable source of natural enzyme inhibitors to combat global health problems including diabetes mellitus and skin disorders.

### 3.4. Principal Component Analysis

Unsupervised principal component analysis and hierarchical clustered analysis were applied to assess the connections between plant parts used on their biological activities. The outcomes are shown in Figure 3. With the percentage of variance of 79.1 and 9% respectively; the first two dimensions that represented a cumulative percentage of 88.1% of variance, seemed sufficient to cover the most information in the dataset. The main dominant biological activities of PC1 were FRAP, DPPH, CUPRAC, Ferrous ion chelating and phosphomolydbdenum while PC2 was dominated by alpha amylase inhibition Figure 3A. Regarding the loading plot, it can be seen that many biological activities were linked with each other Figure 3B. In fact, the greatest positive correlation occurred among tyrosinase and antioxidant properties. The existence of an interesting relationship between antioxidant defense systems and melanogenesis is well documented [76]. In fact, by reacting with toxic ROS result in the restriction of radical chain propagation, eventually preventing the skin from damage. Besides, the cytoprotective antioxidants can be increased by antioxidant molecules thanks to the nuclear accumulation of Nrf2, which is a main transcription factor for the oxidative stress regulation in human skin tissues such as melanocyte, keratinocytes, and dermal fibroblasts [76].

Further, it can be noted the involvement of polyphenols namely hyperoside, (−)-epicatechin, caffeic acid and 2,5 dihydroxybenzoic acid in these activities. Caffeic acid, an important members of hydroxycinnamic acid, (−)-epicatechin and 2,5 dihydroxybenzoic acid are reported to be a good antioxidant with an excellent tyrosinase inhibition properties [77,78,79,80]. In fact, the assays performed on the B16 melanoma cell line showed that caffeic acid can inhibit melanin production by suppressing casein kinase 2 induced phosphorylation of tyrosinase in dose dependent [81]. In addition, a flavanol glycoside, hyperoside is found to be a useful therapeutic agent in the vitiligo management and in the prevention of the oxidative stress induced by reactive oxygen species [82,83]. Regarding the ferrous ion chelating ability it might be predominantly related to the presence of syringic acid, 4-hydroxybenzoic acid, luteolin and eriodictyol.

Looking at the samples plot, a separation between the organs was achieved along PCs, with the leaves and flowers very distant from the three other organs (roots, aerial parts, and stem) (Figure 3C). Afterwards, the hierarchical analysis done on the basis of PCA result, brought out three clusters (Figure 3D). The results obtained in the current study, demonstrate that biological activities of plant differ dramatically from one organ to another. Among the analyzed organs of *A. macrocephanus*, leaves were found to be a promising source, enclosing biomolecules responsible for antioxidant properties and melanoma management ability. This is the result of the difference in quantity and quality of phytocompounds synthesizes in those organs. This quantitative and qualitative difference of phytocompounds is due to the anatomical and morphological structure as well as in several physiological processes that occur in the different organs [84].

## 4. Conclusions

Analysis of phenolic components, and biological potential using antioxidant and enzyme inhibitory assays of *A. macrocephalus* subsp. *finitimus* extracts were conducted for the first time. Twenty-four compounds were identified and quantified in the tested extracts. The levels of these compounds were dependent on the plant parts used. Hyperoside, apigenin, p-coumaric, and ferulic acids were dominant compounds in the extracts. In the connect with chemical profiles, different results were observed for each part in the biological activity assays. Except for metal chelating ability, the extract from leaves exhibited the best biological activities in the performed assays. To sum up, our observations suggest that *A. macrocephalus* subsp. *finitimus* could serve as a prominent source of bioactive agents to combat global health problems caused by oxidative stress. However, further studies are needed to understand the toxic profile, the type of enzyme inhibition and bioavailability of the tested extracts.

## Figures and Tables

**Figure 1 biology-09-00231-f001:**
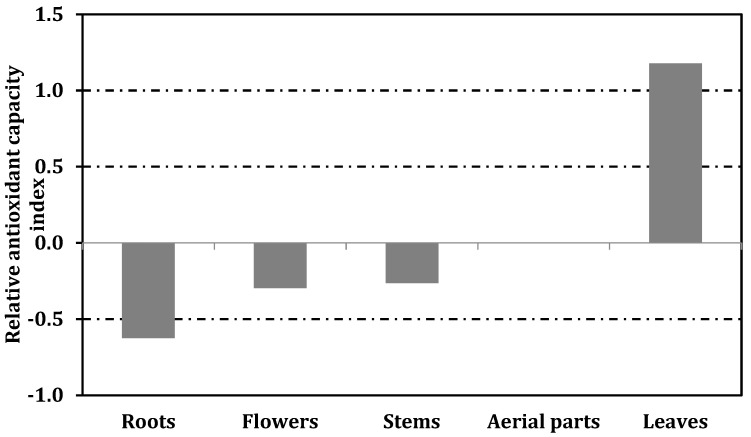
Relative antioxidant capacity index of different parts of *A. macrocephalus* subsp. *finitimus*.

**Figure 2 biology-09-00231-f002:**
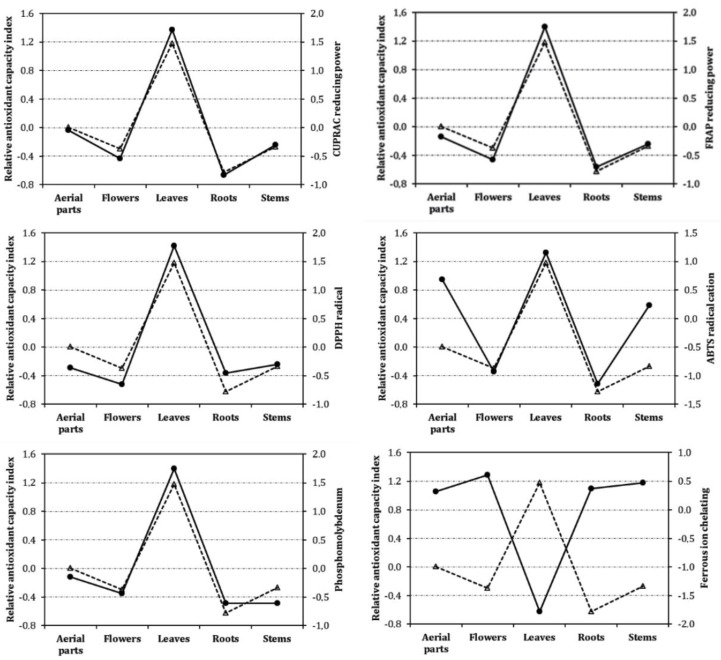
Relative antioxidant capacity index (dashed line with triangle) and antioxidant activity (solid line with circle) of each different part of *A. macrocephalus* subsp. *finitimus.*

**Figure 3 biology-09-00231-f003:**
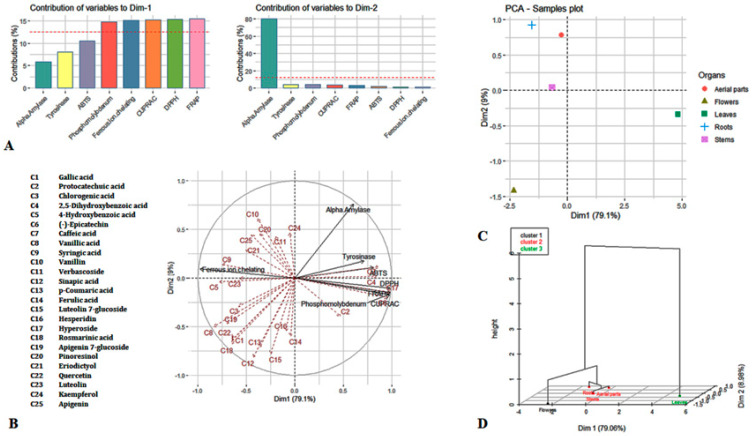
Principle Component Analysis (PCA) and hierarchical clustering analysis. (**A**): Loading plot. (**B**): Contribution of biological activities to each dimension of PCA. (**C**): Samples plot. (**D**): Hierarchical clustering on the fact.

**Table 1 biology-09-00231-t001:** Extraction yield, total phenolic and flavonoid contents of the methanol extracts from different parts of *A. macrocephalus* subsp. *finitimus*
^x^.

Samples	Yield (%)	Total Flavonoids (Mg QE/g Extract)	Total Phenolics (Mg GAE/g Extract)
Aerial parts	10.56	21.06 ± 0.11 ^c^	10.01 ± 0.17 ^b^
Flowers	6.95	29.90 ± 0.95 ^b^	6.96 ± 1.08 ^bc^
Leaves	3.46	39.23 ± 1.64 ^a^	37.68 ± 0.74 ^a^
Roots	17.83	6.03 ± 0.05 ^d^	5.60 ± 0.06 ^c^
Stems	11.78	7.91 ± 0.37 ^d^	8.29 ± 1.13 ^bc^

^x^ Within each column, means sharing the different superscripts (a–d) show comparison between the extracts using Tukey’s test at *p* < 0.05, GAEs and QEs, gallic acid and quercetin equivalents, respectively.

**Table 2 biology-09-00231-t002:** Concentration (µg/g extract) of selected phytochemicals in the methanol extracts from different parts of *A. macrocephalus* subsp. *finitimus*
^x^.

Compounds	Aerial Parts	Flowers	Leaves	Roots	Stems
Gallic acid	2.28 ± 0.03 ^d^	8.76 ± 0.18 ^a^	2.24 ± 0.01 ^d^	4.93 ± 0.12 ^b^	3.03 ± 0.04 ^c^
Protocatechuic acid	14.37 ± 0.08 ^c^	16.80 ± 0.47 ^b^	18.90 ± 0.16 ^a^	13.51 ± 0.64 ^c^	8.06 ± 0.05 ^d^
Chlorogenic acid	8.99 ± 0.24 ^a^	10.29 ± 4.02 ^a^	0.75 ± 0.26 ^b^	3.06 ± 0.69 ^ab^	1.59 ± 0.19 ^b^
2,5-Dihydroxybenzoic acid	9.96 ± 1.74 ^c^	nd	23.14 ± 0.06 ^a^	nd	16.86 ± 1.09 ^b^
4-Hydroxybenzoic acid	12.78 ± 0.16 ^c^	28.83 ± 0.62 ^b^	2.72 ± 0.07 ^e^	31.86 ± 0.76 ^a^	8.13 ± 0.28 ^d^
(−)-Epicatechin	nd	nd	5.60 ± 0.11	nd	nd
Caffeic acid	2.77 ± 0.11 ^b^	nd	3.94 ± 0.03 ^a^	nd	nd
Vanillic acid	23.30 ± 0.23 ^c^	100.53 ± 8.83 ^a^	4.36 ± 0.26 ^d^	57.78 ± 4.27 ^b^	51.53 ± 3.73 ^b^
Syringic acid	6.71 ± 0.03 ^d^	20.89 ± 0.71 ^b^	2.08 ± 0.06 ^e^	33.47 ± 1 ^a^	17.27 ± 1.52 ^c^
Vanillin	3.42 ± 0.11 ^c^	nd	nd	55.68 ± 0.68 ^a^	23.62 ± 0.13 ^b^
Verbascoside	43.48 ± 0.25 ^a^	8.68 ± 0.30 ^b^	1.13 ± 0.08 ^e^	7.29 ± 0.14 ^c^	2.64 ± 0.11 ^d^
Sinapic acid	7.41 ± 0.01 ^b^	52.66 ± 1.36 ^a^	3.41 ± 0.52 ^c^	nd	nd
p-Coumaric acid	76.85 ± 0.47 ^b^	146.78 ± 1.89 ^a^	33.12 ± 0.35 ^c^	6.81 ± 0.60 ^e^	22.12 ± 0.51 ^d^
Ferulic acid	52.79 ± 2.88 ^b^	64.20 ± 2.45 ^a^	37.61 ± 0.30 ^c^	10.53 ± 0.48 ^d^	17.40 ± 0.96 ^d^
Luteolin 7-glucoside	17.63 ± 0.89 ^ab^	28.59 ± 9.16 ^a^	13.45 ± 0.17 ^ab^	6.20 ± 0.75 ^b^	11.35 ± 0.16 ^b^
Hesperidin	7.69 ± 0.04 ^ab^	8.44 ± 1.45 ^a^	7.05 ± 0.03 ^ab^	4.99 ± 0.06 ^b^	9.14 ± 0.75 ^a^
Hyperoside	401.68 ± 6.72 ^b^	90.45 ± 1.39 ^d^	1828.94 ± 21 ^a^	2.90 ± 0.46 ^e^	321.43 ± 7.64 ^c^
Rosmarinic acid	2.10 ± 0.11 ^c^	26.95 ± 3.30 ^a^	1.25 ± 0.02 ^c^	10.46 ± 0.65 ^b^	5.43 ± 0.27 ^bc^
Apigenin 7-glucoside	23.68 ± 0.08 ^c^	63.56 ± 1.94 ^a^	16.85 ± 0.19 ^d^	44.54 ± 0.35 ^b^	20.24 ± 0.04 ^cd^
Pinoresinol	nd	nd	nd	6.40 ± 0.37 ^a^	6.80 ± 0.23 ^a^
Eriodictyol	0.44 ± 0.05 ^c^	2.58 ± 0.36 ^b^	0.30 ± 0.01 ^c^	6.97 ± 0.12 ^a^	0.73 ± 0.01 ^c^
Quercetin	6.47 ± 0.02 ^b^	11.11 ± 0.04 ^a^	3.27 ± 0.06 ^d^	4.58 ± 0.13 ^c^	4.74 ± 0.12 ^c^
Luteolin	32.06 ± 0.45 ^cd^	81.94 ± 0.28 ^b^	32.58 ± 1.07 ^c^	108.11 ± 1.20 ^a^	28.77 ± 0.73 ^d^
Kaempferol	1.58 ± 0.25	nd	nd	nd	nd
Apigenin	29.07 ± 0.50 ^c^	52.33 ± 1.34 ^b^	23 ±0.12 ^c^	181.90 ± 6.23 ^a^	42.81 ± 0.52 ^b^

^x^ Within each row, means sharing the different superscripts (a–d) show comparison between the samples using Tukey’s test at *p* < 0.05. nd, not detected.

**Table 3 biology-09-00231-t003:** Antioxidant activities of standards and the methanol extracts from different parts of *A. macrocephalus* subsp. *finitimus*
^x^.

Assays	Aerial Parts	Flowers	Leaves	Roots	Stems	Trolox	EDTA
Effective concentration (EC_50_: mg/mL)							
Phosphomolybdenum	2.52 ± 0.23 ^c^	2.67 ± 0.08^c^	1.81 ± 0.12 ^b^	2.77 ± 0.03 ^c^	2.77 ± 0.15 ^c^	1.14 ± 0.02 ^a^	-
DPPH radical	9.30 ± 0.20 ^c^	14.20 ± 0.12^e^	2.65 ± 0.05 ^b^	10.54 ± 0.52 ^d^	8.73 ± 0.22 ^c^	0.26 ± 0.02 ^a^	-
ABTS radical cation	1.66 ± 0.01 ^b^	3.35 ± 0.01^d^	1.45 ± 0.04 ^b^	3.88 ± 0.11 ^e^	1.93 ± 0.07 ^c^	0.25 ± 0.02 ^a^	-
CUPRAC reducing power	2.37 ± 0.06 ^c^	3.62 ± 0.16^d^	1.06 ± 0.02 ^b^	5.28 ± 0.39 ^e^	2.90 ± 0.12 ^cd^	0.32 ± 0.03 ^a^	-
FRAP reducing power	1.75 ± 0.02 ^c^	2.46 ± 0.05^d^	0.73 ± 0.01 ^b^	2.83 ± 0.12 ^e^	1.93 ± 0.08 ^c^	0.12 ± 0.02^a^	-
Ferrous ion chelating	1.08 ± 0.01 ^b^	1.03 ± 0.01^b^	1.65 ± 0.05 ^c^	1.07 ± 0.01 ^b^	1.05 ± 0.03 ^b^	-	0.036 ± 0.004 ^a^
Antioxidant activity							
Phosphomolybdenum (mmol TEs/g extract)	1.85 ± 0.17 ^b^	1.73 ± 0.05^b^	2.56 ± 0.17 ^a^	1.67 ± 0.02 ^b^	1.67 ± 0.09 ^b^	-	-
DPPH radical (mg TEs/g extract)	24.97 ± 0.61 ^bc^	15.42 ± 0.15^d^	94.66 ± 1.97 ^a^	21.75 ± 1.21 ^c^	26.80 ± 0.76 ^b^	-	-
ABTS radical cation (mg TEs/g extract)	161.59 ± 0.86 ^b^	78.94 ± 0.01^d^	185.65 ± 5.03 ^a^	67.98 ± 2.01 ^d^	138.54 ± 4.74 ^c^	-	-
CUPRAC reducing power (mg TEs/g extract)	126.91 ± 3.50 ^b^	78.76 ± 3.96^d^	297.74 ± 4.90 ^a^	50.23 ± 4.64 ^e^	101.35 ± 4.66 ^c^	-	-
FRAP reducing power (mg TEs/g extract)	59.81 ± 0.67 ^b^	42.59 ± 0.89^c^	142.69 ± 2.15 ^a^	37.05 ± 1.63 ^c^	54.37 ± 2.15 ^b^	-	-
Ferrous ion chelating (mg EDTAEs/g extract)	66.81 ± 0.36 ^a^	70.11 ± 0.51^a^	43.41 ± 1.39 ^b^	67.45 ± 0.18 ^a^	68.59 ± 2.00 ^a^	-	-

^x^ Within each row, means sharing the different superscripts show comparison between the samples using Tukey’s test at *p* < 0.05. EC_50_ (mg/mL), effective concentration at which the absorbance was 0.5 for reducing power and phosphomolybdenum assays and at which 50% of the DPPH and ABTS radicals were scavenged and the ferrous ion-ferrozine complex were inhibited. EDTA, ethylenediaminetetraacetic acid (disodium salt). “-”, not determined. TEs and EDTAEs, trolox and ethylenediaminetetraacetic acid (disodium salt) equivalents, respectively.

**Table 4 biology-09-00231-t004:** Correlations among phenolic compounds and assays ^x^.

Assays and Compounds	Phosphomolybdenum	DPPH	ABTS	CUPRAC	FRAP	Ferrous Ion Chelating	Tyrosinase	α-Amylase
DPPH	0.974 ^y^							
ABTS	0.717	0.712						
CUPRAC	0.980 ^y^	0.968 ^y^	0.828					
FRAP	0.985 ^y^	0.987 ^y^	0.789	0.995 ^y^				
Ferrous ion chelating	−0.983 ^y^	−0.995 ^y^	−0.678	−0.960 ^y^	−0.980 ^y^			
Tyrosinase	0.549	0.723	0.512	0.606	0.655	−0.671		
α-Amylase	0.497	0.525	0.528	0.477	0.491	−0.552	0.439	
Total flavonoid	0.792	0.657	0.471	0.754	0.725	−0.682	0.009	0.012
Total phenolic	0.992 ^y^	0.992 ^y^	0.727	0.985 ^y^	0.995 ^y^	−0.991 ^y^	0.642	0.478
Hyperoside	0.982 ^y^	0.987 ^y^	0.794	0.995 ^y^	0.999 ^y^	−0.979 ^y^	0.662	0.494

^x^ Data show the Pearson Correlation Coefficients between the parameters. ^y^ Significant at *p* < 0.01.

**Table 5 biology-09-00231-t005:** Enzyme inhibition activities of standards and the methanol extracts from different parts of *A. macrocephalus* subsp. *finitimus*
^x^.

Assays	Aerial Parts	Flowers	Leaves	Roots	Stems	Kojic Acid	Acarbose
Inhibition concentration (IC_50_: mg/mL)							
Tyrosinase inhibition	1.33 ± 0.10 ^cd^	1.41 ± 0.05 ^d^	1.02 ± 0.02 ^b^	1.18 ± 0.01 ^bc^	1.07 ± 0.03 ^b^	0.36 ± 0.04 ^a^	-
α-Amylase inhibition	3.40 ± 0.02 ^b^	4.94 ± 0.15 ^d^	3.36 ± 0.18 ^b^	3.50 ± 0.03 ^b^	4.12 ± 0.22 ^c^	-	1.24 ± 0.06 ^a^
Enzyme inhibitory activities							
Tyrosinase inhibition (mg KAEs/g extracts)	270 ± 20 ^cd^	255 ± 8 ^d^	352 ± 7 ^a^	304 ± 3 ^bc^	336 ± 9 ^ab^	-	-
α-Amylase inhibition (mg ACEs/g extracts)	357 ± 2 ^a^	245 ± 8 ^c^	362 ± 20 ^a^	347 ± 3 ^a^	294 ± 16 ^b^	-	-

^x^ Within each row, means sharing the different superscripts show comparison between the samples using Tukey’s test at *p* < 0.05. IC_50_ (mg/mL), inhibition concentration at which 50% of the α-amylase and tyrosinase activities were inhibited. “-” not determined. ACEs and KAEs, acarbose and kojic acid equivalents, respectively.

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
