# Peer review of "Polyphenol Profile and Biological Activity Comparisons of Different Parts of Astragalus macrocephalus subsp. finitimus from Turkey"

_biology, 2020, doi:10.3390/biology9080231_

Round 1
Reviewer 1 Report
The Authors have tried to improve their manuscript according to all reviewers indications, and now I can recommend the MS for publication on Biology.
Reviewer 2 Report
The quality of this manuscript is considerably improved. The authors took into account reviewers recommendations and they have correct many inaccuracies.
Reviewer 3 Report
Responses to my comments were suitable enough
Several paragraphs have been added as required
Corrections have been done
Aglycones / glycosides or acetylated forms could be studied in a next step of the work, them compared to literature
That is a good basic research for identifying new structures or fractions having biological activities of great interests.
This paper can be published
This manuscript is a resubmission of an earlier submission. The following is a list of the peer review reports and author responses from that submission.
Round 1
Reviewer 1 Report
The manuscript is focusing on some biological activity of almost unknown endemic to Turkey plant from the Astragalus genus – A. macrocephalus subsp. finitimus. The Authors are trying to expand the knowledge in this area and determine polyphenol substances profile, considering them as the main active substances responsible for raw material antioxidant activity. The study is based on an interesting laboratory tests of raw material collected in one localization in Turkey (but not indicating that it was collected from the natural state). It is worth to underline, that the global interest in medicinal plants is raising, similarly as possibility of use its raw material in many different ways. Thus, the manuscript is on important topic and therefore the authors' input to the total knowledge improvement should be very welcome.
The Authors use adequate laboratory methods and described them well in a Material and Methods section as well as in Supplementary materials. In an Introduction section, however, there is no information about a plant species described (its morphology and biology or etnopharmaceutical use). In one statement the Authors wrote only, that :”However, to authors´ best knowledge, very few publications can be found biological properties of Astragalus microcephalus [21-23]”.
In the Material and Methods section there is no information when the raw material was harvested, in which phase of growth and development of plants. Although the Authors divided the A. macrocephalus subsp. finitimus’ raw material on aerial parts (without the specification, whole?), roots and also flowers, stems and leaves, we do not know what was the share of that 3 parts of plants in the whole aboveground plants biomass (maybe it also contained fruit and seeds). It would be very interesting to know that information, because the best activity was in the leaves. In the Material and Methods section I could not find also any information about RACI and its calculations. In the Figure 1 – I could not see the aerial parts. The results presents new and useful information on use possibility of new endemit plant from Astragalus from Turkey.
In conclusion: this manuscript presents an interesting work and the degree of novelty is significant for a large perspective in herbs science and I can recommend the MS for publication on Biology after minor corrections.
Author Response
Reviewer 1: Corrections made have been highlighted in yellow.
The manuscript is focusing on some biological activity of almost unknown endemic to Turkey plant from the Astragalus genus – A. macrocephalus subsp. finitimus. The Authors are trying to expand the knowledge in this area and determine polyphenol substances profile, considering them as the main active substances responsible for raw material antioxidant activity. The study is based on an interesting laboratory tests of raw material collected in one localization in Turkey (but not indicating that it was collected from the natural state). It is worth to underline, that the global interest in medicinal plants is raising, similarly as possibility of use its raw material in many different ways. Thus, the manuscript is on important topic and therefore the authors' input to the total knowledge improvement should be very welcome.
The Authors use adequate laboratory methods and described them well in a Material and Methods section as well as in Supplementary materials. In an Introduction section, however, there is no information about a plant species described (its morphology and biology or etnopharmaceutical use). In one statement the Authors wrote only, that :”However, to authors´ best knowledge, very few publications can be found biological properties of Astragalus microcephalus [21-23]”.
Response: Thanks a lot for your interest. We have been added the informations for the plant.
In the Material and Methods section there is no information when the raw material was harvested, in which phase of growth and development of plants. Although the Authors divided the A. macrocephalus subsp. finitimus’ raw material on aerial parts (without the specification, whole?), roots and also flowers, stems and leaves, we do not know what was the share of that 3 parts of plants in the whole aboveground plants biomass (maybe it also contained fruit and seeds). It would be very interesting to know that information, because the best activity was in the leaves. In the Material and Methods section I could not find also any information about RACI and its calculations. In the Figure 1 – I could not see the aerial parts. The results presents new and useful information on use possibility of new endemit plant from Astragalus from Turkey.
Response: We have been improved this part. A paragraph on the calculation of RACI values ​​has been added to the 2.5. Statistical analysis section. Since the RACI value of aerial parts is 0.003, it does not appear in figure 1.
In conclusion: this manuscript presents an interesting work and the degree of novelty is significant for a large perspective in herbs science and I can recommend the MS for publication on Biology after minor corrections.
Response: Thanks a lot for your positive comments.

Reviewer 2 Report
This manuscript presents the results of the chemical profiling and biological activity of the extracts from the whole aerial parts and different parts of the plant. Total phenolic and total flavonoid content in each extract were determined by colorimetric assays. Concentration of individual phenolic compounds is determined by LC–ESI–MS/MS analysis. Experimental details for this analysis are given in supplementary file and reference [25].
From these attached materials I could not agree that the analysis is correct enough. The mass spectral or UV characteristics of phenolic compounds are not given in the table with analytical data and in addition the data are referred to extracts of Campanula macrostachya (but it could be typing mistake).
The article [25] describing the experimental conditions and used standards refers to olive leaf extract. The selected compounds are probably characteristic for this plant species and are also common to many other species but search in SCOPUS database did not show articles concerned presence of verbascoside or other phenylethanoid glycosides in the genus Astragalus. If the extracts of investigated species Astragalus macrocephalus contains this compound it should be demonstrated by analysis of MS / MS data and in Table 2 in manuscript should be included characteristic ions for all identified compounds and their retention times in particular extracts. The HPLC chromatograms of each extract could be also applied.
The data in Table 1 also do not look quite correct. In most extracts the amount of total flavonoids is higher than that of total phenols especially in the extracts of aerial parts and flowers.
Author Response
Reviewer 2: Corrections made have been highlighted in green
This manuscript presents the results of the chemical profiling and biological activity of the extracts from the whole aerial parts and different parts of the plant. Total phenolic and total flavonoid content in each extract were determined by colorimetric assays. Concentration of individual phenolic compounds is determined by LC–ESI–MS/MS analysis. Experimental details for this analysis are given in supplementary file and reference [25].
From these attached materials I could not agree that the analysis is correct enough. The mass spectral or UV characteristics of phenolic compounds are not given in the table with analytical data and in addition the data are referred to extracts of Campanula macrostachya (but it could be typing mistake).
The article [25] describing the experimental conditions and used standards refers to olive leaf extract. The selected compounds are probably characteristic for this plant species and are also common to many other species but search in SCOPUS database did not show articles concerned presence of verbascoside or other phenylethanoid glycosides in the genus Astragalus. If the extracts of investigated species Astragalus macrocephalus contains this compound it should be demonstrated by analysis of MS / MS data and in Table 2 in manuscript should be included characteristic ions for all identified compounds and their retention times in particular extracts. The HPLC chromatograms of each extract could be also applied.
Response: In Supplementary Material file, ESI–MS/MS Parameters and analytical characteristics for the Analysis of Target Analytes by MRM Negative and Positive Ionization Mode were added to Supplementary Table 1 and LC-ESI–MS/MS chromatograms of the extracts to Supplementary Figure 1.
The data in Table 1 also do not look quite correct. In most extracts the amount of total flavonoids is higher than that of total phenols especially in the extracts of aerial parts and flowers.
Response: Thanks so much for your question. Yes, flavonoids are main group of phenolics. However, we measured the total phenolic and flavonoids content in the tested extracts by using spectrophotometric assays. We used different standards in the assays (gallic acid for phenolics and quercetin for flavonoids). At this point, the total flavonoid content may be higher than total phenolic content in the plant extracts owing to the standards in the tests used. Also, the spectrophotometric assays have several drawbacks. For example, other phytochemicals could be react with the reagents in the assays and then give mistake positive results. Thus, we used further analytical technique, namely LC-MS to confirm the content of phytochemicals.
